# Understanding Sex Differences in Childhood Undernutrition: A Narrative Review

**DOI:** 10.3390/nu14050948

**Published:** 2022-02-23

**Authors:** Susan Thurstans, Charles Opondo, Andrew Seal, Jonathan C. Wells, Tanya Khara, Carmel Dolan, André Briend, Mark Myatt, Michel Garenne, Andrew Mertens, Rebecca Sear, Marko Kerac

**Affiliations:** 1Department of Population Health, London School of Hygiene and Tropical Medicine, London WC1E 7HT, UK; rebecca.sear@lshtm.ac.uk (R.S.); marko.kerac@lshtm.ac.uk (M.K.); 2Department of Medical Statistics, London School of Hygiene and Tropical Medicine, London WC1E 7HT, UK; charles.opondo@lshtm.ac.uk; 3National Perinatal Epidemiology Unit, Nuffield Department of Population Health, University of Oxford, Oxford OX3 7LF, UK; 4UCL Institute for Global Health, London WC1E 6BT, UK; a.seal@ucl.ac.uk; 5Population, Policy and Practice Research and Teaching Department, UCL Great Ormond Street Institute of Child Health, London WC1N 1EH, UK; jonathan.wells@ucl.ac.uk; 6Emergency Nutrition Network, Oxford OX5 2DN, UK; tanya@ennonline.net (T.K.); carmel@n4d.group (C.D.); 7Center for Child Health Research, School of Medicine, Tampere University, 33520 Tampere, Finland; andre.briend@gmail.com; 8Department of Nutrition, Exercise and Sports, University of Copenhagen, DK-2200 Copenhagen, Denmark; 9Brixton Health, Llwyngwril, Gwynedd LL37 2JD, Wales, UK; mark@brixtonhealth.com; 10Institut de Recherche pour le Développement, UMI Résiliences, 93140 Bondy, France; mgarenne@hotmail.com; 11Institut Pasteur, Epidémiologie des Maladies Emergentes, 75015 Paris, France; 12Senior Fellow, FERDI, Université d’Auvergne, 63000 Clermont-Ferrand, France; 13MRC/Wits Rural Public Health and Health Transitions Research Unit, School of Public Health, Faculty of Health Sciences, University of the Witwatersrand, Johannesburg 2193, South Africa; 14School of Public Health, University of California, Berkeley, CA 94720-7360, USA; amertens@berkeley.edu; 15Maternal, Adolescent, Reproductive & Child Health Centre (MARCH), London School of Hygiene & Tropical Medicine, London WC1E 7HT, UK

**Keywords:** undernutrition, sex, age

## Abstract

Complementing a recent systematic review and meta-analysis which showed that boys are more likely to be wasted, stunted, and underweight than girls, we conducted a narrative review to explore which early life mechanisms might underlie these sex differences. We addressed different themes, including maternal and newborn characteristics, immunology and endocrinology, evolutionary biology, care practices, and anthropometric indices to explore potential sources of sex differences in child undernutrition. Our review found that the evidence on why sex differences occur is limited but that a complex interaction of social, environmental, and genetic factors likely underlies these differences throughout the life cycle. Despite their bigger size at birth and during infancy, in conditions of food deprivation, boys experience more undernutrition from as early as the foetal period. Differences appear to be more pronounced in more severe presentations of undernutrition and in more socioeconomically deprived contexts. Boys are more vulnerable to infectious disease, and differing immune and endocrine systems appear to explain some of this disadvantage. Limited evidence also suggests that different sociological factors and care practices might exert influence and have the potential to exacerbate or reverse observed differences. Further research is needed to better understand sex differences in undernutrition and the implications of these for child outcomes and prevention and treatment programming.

## 1. Introduction

The reduction of childhood undernutrition in the form of wasting, stunting, and being underweight is key to the World Health Assembly targets and Sustainable Development Goals and is the specific focus of Goal 2, as well as being relevant to many others. Better understanding of the underlying aetiology and risk factors is key to successful prevention and management and the achievement of these goals.

In a recent systematic review and meta-analysis on sex differences in undernutrition in children aged under 5 years, [1] we found that in most settings, boys are more likely to be wasted (pooled OR 1.26, 95% CI 1.13 to 1.40), stunted (pooled OR 1.29, 95% CI 1.22 to 1.37), and underweight (pooled OR 1.14, 95% CI 1.02 to 1.26) than girls. There were regional differences, with the increase in risk being less pronounced for boys in South Asia, suggesting that social, genetic, or environmental factors may affect sex differences in nutritional status. 

This finding that boys are more likely to be undernourished than girls is supported by a number of other studies. A pooled analysis of 35 longitudinal cohorts from 15 LMICs showed male sex to be a predictor of both wasting and stunting [2]. Several studies exploring concurrent wasting and stunting have also shown that, overall, boys are more likely to be affected than girls; these include population-level data [3,4,5,6,7], some of which contain multiple data sets, and Severe Acute Malnutrition (SAM) treatment programme data [8,9,10]. A recent analysis of DHS data from Africa explored sex differences in undernutrition and found that though differences were small, overall, boys were more susceptible to undernutrition than girls. The biggest differences were found in children who were concurrently wasted and stunted. Within this group, sex differences were more than the sum of sex differences in individual stunting and wasting, revealing complex layers of vulnerabilities [7]. Our review also found that sex differences tend not to be systematically reported or sometimes even recognised in studies of child undernutrition and that, where they are, explanations provided for these differences are often conjectural. Despite an abundance of literature related to sex differences in childhood in fields such as neonatal health or general morbidity and mortality patterns, to date, we have not identified a study which has specifically focused on the potential causes of sex-differences in undernutrition.

Understanding the possible origins, pathways, and consequences of sex differences in undernutrition is important for practitioners in the nutrition community and will help to refine what implications there may be for policy and practice. This paper complements our meta-analysis by presenting an in-depth exploration of potential reasons for the observed sex differences in undernutrition and considering what the pathways behind those differences may be. The aim is to synthesise data from relevant public health and biomedical domains to deepen our understanding of sex differences in undernutrition and consider their potential relevance to policy makers and practitioners in the sector. 

## 2. Materials and Methods

This is a narrative review complementing our systematic review [1], in which we explore some of the themes identified for explaining sex differences in undernutrition in depth. The original systematic review followed the Preferred Reporting Items for Systematic reviews and Meta-Analyses (PRISMA) guidelines [11]. The broad nature of the original protocol led to the identification of studies that were highly heterogeneous in the design and outcomes being studied, and so the study was divided into two parts. The first focuses on describing the sex-specific prevalences in undernutrition and how these varied by age and geographic region, and the second (this paper) focuses on possible explanations, exploring genetic, physiological, environmental, and sociological factors for these sex differences. 

The original search strategy is reported in an earlier publication [1] and was designed to capture the concepts of malnutrition, sex, and gender. Medline, Embase, Global health, Popline, and Cochrane databases were searched. In this narrative review, we also included studies that were considered relevant to the subject of sex differences but were not eligible for the systematic review, due to the absence of extractable prevalence data. We further searched the references included in the studies identified through the original search. Here, therefore, we have included relevant studies which discussed possible explanations or presented evidence for the possible causes of sex differences in undernutrition.

We also linked DHS data on the prevalence of stunting to FAO data on food security scores at the country level to graphically examine the relationship between the country-level prevalence of stunting and food insecurity using Stata v15 (Stata Corp., College Station, TX, USA). Countries were included in the analysis if data were available both from the DHS stat compiler and in the FAO table of food security scores.

Throughout this paper, we use the following WHO definitions: Low birthweight (LBW) refers to a birthweight < 2500 g; wasting is defined as a weight-for-height z-score < −2 or a MUAC under 125 mm; stunting and underweight are defined, respectively, as a height-for-age z-score < −2 and a weight-for-age z-score < −2.

## 3. Results: Potential Explanatory Factors for Sex Differences

### 3.1. Maternal and Newborn Factors

Differences in birth outcomes between girls and boys have been well recognised for many years. In the case of undernutrition, the fact that the greatest sex differences are found in early ages suggests that differences might originate, at least in part, in-utero [7]. Male sex has been identified as an independent risk factor for adverse pregnancy, maternal, and foetal outcomes [12,13]. Male foetuses are at increased risk of poor outcomes when compared with female foetuses and are more likely to experience complications, such as placental insufficiency, infections, and pre-term delivery [12,14,15]. Males are also less likely to survive premature birth with immature lung development, including the later development of surfactant [16]. It is estimated that a newborn female is physiologically similar to a 4–6 week old male [17], suggesting that, at birth, females are already more developed and therefore more able to withstand adverse conditions.

During the foetal period, the health and nutritional status of the mother can impact foetal growth and development and pregnancy outcomes. Poor nutritional status is associated with birth complications and poor outcomes [18,19]. Evidence has shown that the first 1000 days between conception and a child’s second birthday is a critical period in which the foundations for long-term growth and development are determined [20]. The high levels of maternal undernutrition [21] in resource-limited settings have implications for meeting the needs of both the mother and the foetus. In relation to sex differences, this leads us to question if, in resource-poor settings, maternal nutritional status may have a bigger impact on infant boys than girls.

Maternal nutrition plays a pivotal role in the regulation of placental-foetal development. The role of the placenta is to provide oxygen and nutrients and facilitate the exchange of waste between the maternal and foetal circulations. Its size and function determine pre-natal growth and the infant’s growth trajectory and are directly correlated. When maternal undernutrition occurs, changes in the structure of the placenta ensue, reflected in placental weight, morphology, vascular development, and transport function for amino acids, resulting in an altered nutrient supply to the foetus [22] and exacerbating the state of biological competition between a mother and foetus [19,23]. In this instance, the risk of undernutrition in the foetus is increased, [24] potentially affecting the lifelong health and productivity of offspring [22]. This might therefore suggest that boys may be in greater competition with their mothers than girls. Wells [24] suggests that in good environmental conditions, mothers physiologically invest proportionally more in a male than in a female foetus during the second half of pregnancy, resulting in a higher deposition of lean tissue. 

Foetal growth in girls and boys differs from an early stage. Male foetuses have, on average, been found to be larger than female ones from the 8th to 12th week, suggesting genetic mechanisms underlying sex differences in foetal size [25]. Erikson et al. [26] describe sex differences in growth during gestation and explore the reasons and impacts of this. They suggest that boys grow faster in the womb than girls from the early stages of gestation (even before implantation). The research shows that boys’ placentas may be more efficient than girls’ placentas, as boys are usually longer in length than girls at any placental weight. They also suggest, however, that boys’ placentas might have less reserve capacity (ability to transfer oxygen and nutrients to the foetus), increasing their vulnerability to undernutrition. Though they found both mean birthweight and placental weight to be higher in boys, when placental measurements were expressed as a ratio to birthweight, the values for placental measurements were lower in boys. This means that in situations where there is not a free flow of nutrients from the mother and where they need to be sustained by the transfer capacity of the placenta, there is less of this reserve available to a boy than to a girl of the same weight. Therefore, boys’ bigger sizes and faster growth in-utero means they are at increased risk of becoming undernourished before birth. The same study also suggests that during development in the womb, boys are more responsive to a mother’s gestational diet than girls, whose development seems to more closely reflect the mother’s long-term nutrition and metabolic profile, a finding supported in other studies [27,28,29,30].

Recent longitudinal evidence from Nepal [31] supports these ideas. An analysis of data from a randomised controlled trial explored differences in maternal and early child nutritional status by offspring sex. Overall, the authors found that in a population with high levels of maternal undernutrition, there was minimal evidence from nutritional markers that mothers of sons could meet the extra absolute energy costs required for nourishing their sons. Sons, therefore, showed higher rates of stunting and lower head circumference z-scores in early life. However, the sex difference in undernutrition also decreased in favour of boys in the trial arm receiving food supplementation. This suggests that if the nutritional constraint on the mother is relaxed, boys capture a greater benefit than girls and can recover some of their ‘lost’ growth. 

At birth, differences in growth mean that low birth weight (LBW), an indicator strongly linked to mortality, is higher in girls [32]. LBW is defined by an absolute weight (<2500 g) that does not take account of sex. Evidence has shown that despite the higher incidence of LBW in girls, LBW boys experience higher mortality than LBW girls [33]. In other words, it appears that smaller body size and greater adiposity from an early age favour girls in conditions of food deprivation [24,34], whilst a greater dependence on immediate maternal nutrient intake and faster growth may leave boys more vulnerable when later food shortages occur. 

Finally, limited evidence suggests that there are also potential nutritional consequences for women carrying male foetuses. A small US study examined energy intake among pregnant women carrying boys compared with girls. Their study of 244 pregnant women showed that women carrying a male foetus had a higher energy intake than women carrying females (mean change in energy 796.2 kJ/day (95% CI 8.9–1583.4, *p* 0.05)). In agreement with the finding of higher rates of male foetal growth, the authors argue that their findings support the hypothesis that women carrying male foetuses may have higher energy requirements, and therefore, male foetuses may be more susceptible to energy restriction [35].

### 3.2. Endocrine/Immune Factors

Overall morbidity and mortality rates are higher throughout life in male humans compared with females [36]. Boys are generally more susceptible to infectious disease than girls, with some exceptions, notably measles, whooping cough, and tuberculosis [37]. The stronger immune response and capacity of girls for producing antibodies is one explanation for this enhanced immune reactivity [38,39]. Underlying mechanisms for these sex differences are complex and include differences in endocrine and genetic effects on the immune system and physiology. The cycle of undernutrition and infection has been well documented [40,41]. An inadequate dietary intake leads to weight loss, growth faltering, lowered immunity, and mucosal damage, increasing the risk of disease while simultaneously resulting in increased energy and micronutrient requirements for an immune response. Infections further predispose to undernutrition through appetite loss, malabsorption, and altered metabolism and, in turn, inadequate dietary intake and further appetite loss. Hence, if males are more likely to be affected by infections, they are more likely to be affected by undernutrition. Hack et al. [42] found that LBW females achieved greater catch-up growth than LBW males and suggested this may be partly related to a lower incidence of neonatal and child infections, highlighting the importance of addressing infectious disease.

Hormonal systems differ between boys and girls, and the interactions between sex hormones and environmental factors have consequences for energy consumption, nutritional requirements, and vulnerability to infectious and noncommunicable diseases. [43,44] Testosterone, luteinizing hormone (LH), and follicle stimulating hormones (FSH) have been identified as hormones which differ between boys and girls and which might impact susceptibility to infections [45]. Leptin has also been identified as potentially playing a role in different sex responses to undernutrition. Leptin is the “satiety” hormone, responsible for the regulation of food intake and energy expenditure, and it is produced by white adipose tissue, which is higher in girls. When leptin levels are low, a feedback loop with the “hunger” hormone ghrelin is triggered, resulting in a hunger response and increased catabolism of the body’s energy stores [46]. Leptin plays an important role in the generation and maintenance of immune responses [47] and has the capacity to both enhance and impede immune functions, boosting immune functions such as inflammatory cytokine production in macrophages, granulocyte chemotaxis, and increased Th17 proliferation [48].

Leptin is detectable in foetal cord blood from as early as 18 weeks of gestation, dramatically increasing after 34 weeks. A number of studies have demonstrated higher levels of leptin in girls in the last weeks of gestation and in the neonatal period [49,50] These higher levels of fat and leptin in early life for females should, in theory, increase immune protection and resistance to infections that slow growth. In newborns, the serum concentration of leptin is positively correlated with body weight and body mass and both correlated with and produced by fat mass, and it is higher in females than males [49,51]. In a sample of 82 newborns, serum leptin concentrations were found to be significantly lower in males compared with females (mean 15.3, SD 15.6 ng/mL, and range 2.0 to 79.3 ng/mL versus mean 25.0, SD 18.0 ng/mL, and range 2.1 to 84.5 ng/mL, respectively; *p*-value for test of difference = 0.011) [52].

In the case of prolonged nutritional deprivation, leptin production is suppressed by decreased energy-intake, decreasing blood insulin and possibly IGF-I concentrations [50]. A Ugandan study [53] examined the hormonal and metabolic profiles of children admitted for treatment for acute malnutrition in Uganda. As with previous studies, [54] they demonstrated the use of fatty acids as an energy source in acute malnutrition, suggesting that fatty acid metabolism plays a central role in adaptation to childhood malnutrition. They found that low baseline levels of leptin were associated with the highest risk of mortality during treatment for children with acute malnutrition. This suggests that in conditions of deprivation, girls are able to draw on their fat to increase their survival, while males will have less available due to their greater investment of energy in maintaining lean mass. 

### 3.3. Age

Studies have noted that sex differences in undernutrition might be moderated by age [5,7]. Our systematic review showed that the male disadvantage was greater among younger children. In the case of concurrent wasting and stunting, sex ratios change with age, with a higher susceptibility for boys up to 30 months, which then disappears [7,46]. 

Other studies we reviewed touched upon how the relationship between age and undernutrition differs by sex, with some inconsistencies. Adair and Guilkey [55] studied children in the Philippines and found that males were more likely to become stunted in the first year of life, but females were at a higher risk in the second year. In contrast, Bork and Diallo [56] studied children in Senegal and found a significant interaction between age and sex, in that the deficit in boys compared with that in girls increased between the first and second years of life. Moestue [57] examined the height-for-age growth curves of children aged 6–23 months in Bangladesh using the 2006 WHO growth standards. She found girls to have higher WHO z-scores than boys, with the greatest difference in the 6–11-month age group. However, she found that girls’ nutritional advantage diminishes with age, with girls’ Z-scores decreasing faster than boys’.

The lack of consistency across studies suggests heterogeneity in how the relationship between sex and undernutrition is affected by age. The current weight of evidence suggests that boys are more likely to be malnourished in early life but that the male disadvantage might lessen with age. This is further supported by a recent analysis of linear growth in 87 low-income countries, which showed that up to around 30 months, boys are more likely than girls to present growth faltering when compared with international growth standards, but this difference was reduced between 30–45 months, with reversal of the gap in some countries [58]. Understanding whether and how the relationship between sex and undernutrition varies by the child’s age across different contexts might provide some clues to why sex differences in undernutrition exist. 

### 3.4. Evolutionary Explanations

Theories of evolutionary biology are compatible with both the physiological and behavioural explanations discussed in this paper. This may help provide explanations for the male disadvantage, whereby even before birth, males are more vulnerable than females across several domains [59]. Greater biological vulnerability of males is common across mammalian species, and a recent paper [60] suggested that this might be linked to the nature of sex chromosomes. In mammals, males are the heterogametic sex (meaning their sex chromosomes are different (XY)), and females are the homogametic sex (meaning their sex chromosomes are the same (XX)). The ‘unguarded X hypothesis’ suggests that the reduced size of the Y chromosome exposes deleterious mutations on the X chromosome, leading to greater male vulnerability and to greater mortality risk. Supporting this hypothesis is the observation that in other species, such as birds, where males are the homogametic sex, it is males rather than females who have a survival advantage [60].

As well as this mechanistic explanation for the generally greater vulnerability of males, there is also a hypothesis from evolutionary biology which provides an ultimate, functional explanation for why parents may invest less in boys than girls in poor ecological conditions (e.g., under nutritional stress), which could help explain higher rates of male undernutrition in resource-stressed populations. The Trivers–Willard argument (Trivers and Willard, 1973) states that, in mammals, females almost always reproduce, while males have to compete for reproduction. Males in better conditions will be more successful at this competition than males in poor conditions. In this situation, evolutionary theory predicts that in poor conditions females will maximize their reproductive success by giving birth to more daughters, while in good conditions, they will produce more sons [61].

The same argument has been proposed to explain male disadvantage in both undernutrition of young children and under-five mortality [Wells 2000, 2016]. Here, the argument is about parental investment: in poor conditions, it is suggested that mothers biologically invest more in girls, leading to excess male under-nutrition. This hypothesis has stimulated considerable research, and some lines of evidence do provide support [62]. For example, women in Ethiopia who had recently given birth to sons had a better nutritional status than those who had given birth to daughters [63]. Overall, however, evidence is rather mixed. Further, as discussed in the next section, while parents do sometimes treat sons and daughters differently, such parental investment behaviours are complex and not simply explained by the Trivers–Willard hypothesis.

### 3.5. Infant and Young Child Feeding and Care Practices

The way in which an infant is fed has consequences for their growth and development [64]. In breastfed infants, there is some evidence to suggest that the quality and quantity of milk that a mother produces or an infant takes in can vary between male and female infants. A multi-country study [65] exploring human milk intake found that male infants had human milk intakes which were 5% greater than those of female infants, although the data does not refer to exclusively breastfed infants. In a small separate US-based study [66], the breastmilk of mothers with male infants was found to have a 25% greater energy content than that of mothers of female infants. In contrast, a study of 103 Filipino mothers found no difference in milk composition of high- and low-income mothers, no difference in nursing frequency, and no difference by infant sex [67]. Important to note here is that breastmilk context is known to be highly variable from day to day and between feeds. A more recent study [68] found that the composition of milk in animal offspring was dependent on sex and suggested that adapting early nutrition intake might be one mechanism to maximise health protection and development for both sexes. The authors note, however, that the evidence for sex-specificity in human milk composition is currently limited and conflicting, and further investigation would be required to draw useful conclusions.

Some evidence also suggests that the protective effect of breastmilk in infants experiencing acute respiratory infection might differ between sexes [69]. Multivariable analysis found breastfeeding to be protective against pneumonia and hospitalisation in girls but not boys, although non-breastfed girls were at increased risk of severe acute respiratory disease compared with males. These results should be interpreted with caution, however, given the small sample size and the fact that the study did not differentiate between exclusive and non-exclusive breastfeeding. 

Parental investment behaviours differ by context. A number of studies have shown that boys often receive complementary food earlier than girls, either due to boys being perceived as hungrier or because breastmilk was seen as inferior to complementary foods, and boys were prioritised for what was seen to be the superior option. This might lead to increased risk of infection. A longitudinal study in Senegal [56] explored the relationship between sex, nutritional status, and infant and young-child feeding (IYCF). They found that the stunting prevalence was higher for boys than girls in all age groups and that the mean HAZ score was lower for boys than girls in all age groups. The analysis showed sex differences in early initiation of complementary feeding, particularly in the 2–3-month age group. Boys were more likely to have consumed complementary foods (CF) in the past 24 h than girls. These differences were relatively modest and no longer apparent at 4–5 months. The authors note, however, that maternal motivations for introducing CF earlier than recommended in this setting included “a small weak infant” alongside perceived breast milk insufficiency; therefore, the possibility that boys were at greater risk of early CF because they already had a lower mean HAZ than girls cannot be excluded. Similarly, a study using ethnographic interviews in Guatemala [70] found that mothers reported that male infants were hungrier and not as satisfied with breastfeeding alone compared with girls. As a result, boys were introduced earlier to complementary feeds than girls. Demographic and Health Survey (DHS) anthropometric data analysed in the same study showed a height-for-age difference in children between 6 and 17 months of 1.61 cm in favour of girls (*p* < 0.001).

### 3.6. Sex and Socioeconomic Status

We identified some studies which show that sex differences, with increased risk among boys, are more pronounced among lower socio-economic groups. A study of 16 demographic health surveys in Sub-Saharan Africa [71] showed that sex differences in stunting were more pronounced in the lower socio-economic strata. This finding, however, was not uniform, and the authors called for more research to confirm findings. Other evidence might also point to this being a factor. Concurrent wasting and stunting has been found to be significantly higher both in males and in fragile and conflict-affected settings (FCS), [3] suggesting that in areas of deprivation and lower socio-economic conditions, the difference between girls and boys is more pronounced. 

We linked stunting prevalence from DHS surveys with country-level food FAO security scores (see Figure 1) to graphically examine the relationship between the two. The results show that in most countries, as wealth decreases, the prevalence of stunting increases and that the prevalence of stunting is higher among males compared with females. There was strong evidence of a correlation between food insecurity and median prevalence of stunting (correlation coefficient −0.65, *p* < 0.001). The trend in the difference between male and female prevalence also increases as wealth decreases, suggesting the wealthier a country, the less pronounced the difference between boys and girls. Although the pattern is not uniform, and the comparison would benefit from more in-depth analysis, it does suggest that addressing inequality in socio-economic status might also help to reduce sex differences in undernutrition. 

In contrast to the above findings, however, a recent study exploring sex differences and mortality patterns showed that whilst the overall prevalence of undernutrition declined in tandem with decreasing mortality in the population, sex differences in undernutrition increased with declining mortality [7]. This suggests that girls might benefit more than boys from general population health improvements at some levels. 

### 3.7. Gender Perceptions

In addition to feeding and care practices, the social and economic circumstances in which families live and in which children are raised exert influences on health and nutrition outcomes through aspects such as how children are fed, what services they can access, and the wider health and disease environment to which they are exposed. This might explain some of the differences observed between data from Africa and Asia described throughout this review. 

The roles that boys and girls assume within a community and the values attached to them might affect both the nutritional inputs that are made available to them and their exposure to disease and infection. For example, gender roles can influence where boys and girls spend their time and, in turn, the environment to which they are exposed and their access to food. In parts of sub-Saharan Africa, there is often a high value placed on girls because of their role in agriculture and due to the fact that they are seen as an investment, particularly in lower socio-economic groups, and a form of social security for parents [72,73,74]. Likewise, in early childhood, some studies suggest that the time girl children spend around the home might give them an advantage in the attention they receive from parents and increased access to food during food preparation [75]. Male children, on the other hand, might spend more time out of the house, playing with other male children, resulting in greater energy expenditure and exposure to environmental risks and sources of infection [73,75,76]. Despite these differences, a cross sectional study of African DHS data [77] found that overall, African mothers were unlikely to treat male and female infants differently; however, more in-depth mixed methods studies would be required to explore this fully.

In Asian studies, birth order and sibling sex have been found to play a role in the nutritional status of girls and boys, which might explain the differences we identified in our meta-analysis. In a study [78] observing gender differentials in childhood feeding, immunization, treatment seeking, and the nutritional status of children in Northern India, results showed that the extent of gender differentials depended on the birth order of the index child and the sex composition of older living siblings. Girls were less likely to have received solid/semi-solid food during the last 24 h, reflecting the opinion that solid/semi-solid foods are considered to be more valuable compared to liquids and breastfeeding. In contrast, a separate analysis [79] explored breastfeeding duration in India using national family health service data from 1992, 1998, and 2005 and found that girls are breastfed for a shorter duration than boys over concern of the contraceptive effects. They attributed this to both birth order and son preference, showing that duration of breastfeeding increased with birth order (younger children breastfed for longer) and was lowest for daughters, particularly those with no older brother whose parents were still trying for a son. One study documenting behaviours during famine noted that in Bangladesh, boys are given preferential treatment by parents, and girls are more likely to die [80]. These findings are in contrast to other studies which show that females are more likely to survive famine and have better long-term outcomes [81].

### 3.8. Indicators of Undernutrition

How undernutrition is assessed and defined has potential consequences for understanding how sex differences manifest in undernutrition. Weight and height measurements and sex-specific weight-for-height Z-scores are widely used to identify wasting. Unadjusted middle-upper-arm circumference (MUAC) measures are also widely used and have been shown to identify children with a high risk of mortality [82], providing a low cost, easy alternative, which can be used by all levels of health care professionals and mothers themselves [83]. 

The 2006 WHO growth standards [84] describe the growth of a “gold standard” reference group of children, from six different countries, all growing up in optimal environmental conditions and breastfed according to WHO recommendations. This reference data was used to generate Z-scores (standard deviation scores), against which the growth and size of other children in other settings could be compared. Though sex-specific male and female growth references are always used by the WHO, what is not certain is whether sex differences observed in this original reference population are representative of sex differences in all other populations, particularly those living in situations of deprivation. In other words, it is unclear if a specific z-score (<−3, for example) in a girl or boy or the same age corresponds to the same physiological impact in both sexes and how the distribution of fat and fat-free mass affects this. 

The use of MUAC, on the other hand, is based on a single cut-off point for girls and boys. The fact that boys are bigger in absolute terms and have higher energy needs to grow along given centile lines could mean that the same absolute supply of energy would only meet the requirements of a thinner arm in a boy compared to a girl, meaning boys could potentially have a predisposition to a thinner MUAC, once again as a consequence of their slightly larger size. Keeping the MUAC cut-off the same for boys and girls would then see that susceptibility expressed. It has been suggested that using a single cut-off point for MUAC may result in the overestimation of wasting in girls and the underestimation of wasting in boys [85]. Rasmussen et al. [86] compared MUAC with the MUAC Z-score as a predictor of mortality in a cohort of children in Guinea Bissau. As would be expected, they found that MUAC classified more girls and young children as moderately malnourished compared with the MUAC z-score (since z-score tables are sex-specific) but that sensitivity varied across the time-period, sex, and age groups. Overall, they found no difference in the performance of MUAC and MUAC z-score as predictors for mortality. A further analysis of mortality outcomes for older boys and girls using MUAC in different settings would be helpful to better understand risks. 

## 4. Possible Implications of Sex Differences for Undernutrition Programming and Policy

### 4.1. Sex Differences in Treatment Outcomes and Mortality Implications

Though evidence clearly shows a higher risk of wasting, stunting, and being underweight for boys compared with girls, a more detailed analysis is needed to better understand the implications of these differences in relation to health and nutrition outcomes. For example, do differences in incidence and outcome follow the same direction? Evidence around diarrhoeal disease in children aged 1–5 suggests that despite slightly higher incidence rates for males, cause-specific mortality is higher amongst females, perhaps due to health-seeking behaviours [76]. Similarly, in a study of children in Senegal with concurrent wasting and stunting, sex differences in mortality were not significant after controlling for stunting and wasting [45].

In relation to treatment, evidence is limited, but it might indicate longer recovery time in boys. Data from a malnutrition treatment programme in Uganda showed that females had an increased probability of recovery compared with males, though the difference was not significant [10]. Similarly, a meta-analysis of RCTs providing small quantity lipid nutrient supplements (SQ-LNS) found that the effects of SQ-LNS on stunting, wasting, low MUAC, and small head size were greater among girls than among boys [87]. Girls were found to have a better growth status than boys. The authors suggest that the difference probably reflects a greater potential to respond to nutritional supplementation among girls.

### 4.2. The Policy Environment

Policy discussions and directions in the nutrition sector have rightly recognised and highlighted the importance of gender; however, the idea of sex differences is not at present widely recognised, and it is, in some cases, misrepresented. For example, our personal communication with practitioners at a 2019 international nutrition conference suggests that where male disadvantage is shown, data quality is often challenged, as it does not fit prior assumptions. In many areas, women and girls are often identified as more vulnerable, as they may face barriers to gaining equal access to education, health care, work, and representation in both political and economic decision making. When it comes to nutrition, women and girls are also identified as the more vulnerable of the two sexes [88]. The international focus on gender concerns is well-justified. An analysis of data on mortality, wasting, and stunting from 96 countries showed that independent of gross domestic product (GDP), greater societal gender inequality is associated across countries with greater child undernutrition for both girls and boys in the next generation [89]. However, to focus solely on women and girls is not compatible with the promotion of gender equity in health, and as illustrated by our findings, it would be incompatible with public health ethics due to the requirement to focus on all groups who are at risk. Our recent systematic review and meta-analysis highlighted the importance of disaggregation of sex data in nutrition treatment programmes so that admissions can be assessed to see if they reflect the national sex distribution in the undernutrition burden between boys and girls [1]. Although there is no indication that males and females require different nutrition interventions, the recognition of sex differences in undernutrition at the policy level might support disaggregation and better understanding of data at the programme level.

## 5. Discussion

The concept of a sex disadvantage in neonatal and infant health is well described within some health fields, but it is less well understood in the field of nutrition. We reviewed evidence which suggests that there are different stages in the maternal and child lifecycle at which sex differences in nutritional status can manifest, although the strength of this evidence is still limited. 

Boys have higher odds of being undernourished during early childhood in low resource settings, and these differences appear as early as the foetal period. Overall, differences are small but do appear to be more pronounced in more severe presentations of undernutrition, i.e., concurrent wasting and stunting and in more socio-economically deprived contexts with more severe levels of fragility and deprivation. Though genetic vulnerability might initially explain this, the sex differences observed between contexts suggests that a complex interaction of social, environmental, and genetic factors underlies these differences throughout the life cycle. 

This review has demonstrated a number of possible explanations for sex differences in undernutrition, but there is more to be done in terms of fully understanding some of the complexities related to the risks of wasting, stunting, and being underweight for boys and girls. Further research is warranted to understand if sex differences impact treatment outcomes, cognitive development, long-term morbidity, and mortality risk in order to understand what the implications for policy and practice, if any, might be. Likewise, exploration of the different indices, such as MUAC, WHZ, WAZ, HAZ, and LBW, used to define undernutrition is needed. This would help to better understand the points at which differences might occur as children fall further below the reference population to determine if sex differences in outcomes are more pronounced at more severe degrees of undernutrition. 

This review also highlights how complex and, at times, conflicting the evidence is. One of the main challenges encountered was that much of the work we reviewed explored sex differences as a secondary finding, rather than as the main focus of investigation, meaning that explanatory or confounding factors were not always fully accounted for. Further investigation is needed to explore, in detail, the pathways and drivers of sex differences, such as genetics, socio-economic status, infectious diseases, environmental exposures, social preferences for gender and associated practices, and geographical patterns. Understanding how these interact with other factors to impact sex differences at different points in a child’s life will help to determine what actions may be appropriate from a programme or policy standpoint. Likewise, a better understanding of how these complex factors interact to result in, enhance, or reverse sex differences is needed and of what the implications of these may be for treatment and prevention programming. 

The operational implications of these findings are limited at present but do offer some possible explanations for observed sex differences in both survey and treatment data. They also highlight the importance of addressing the drivers of undernutrition, such as socio-economic inequality, and the prevention and treatment of infectious diseases. Finally, they highlight the importance of the continued collection, disaggregation, and analysis of data by age and sex to both identify and target prevention and treatment interventions toward vulnerable children.

## Figures and Tables

**Figure 1 nutrients-14-00948-f001:**
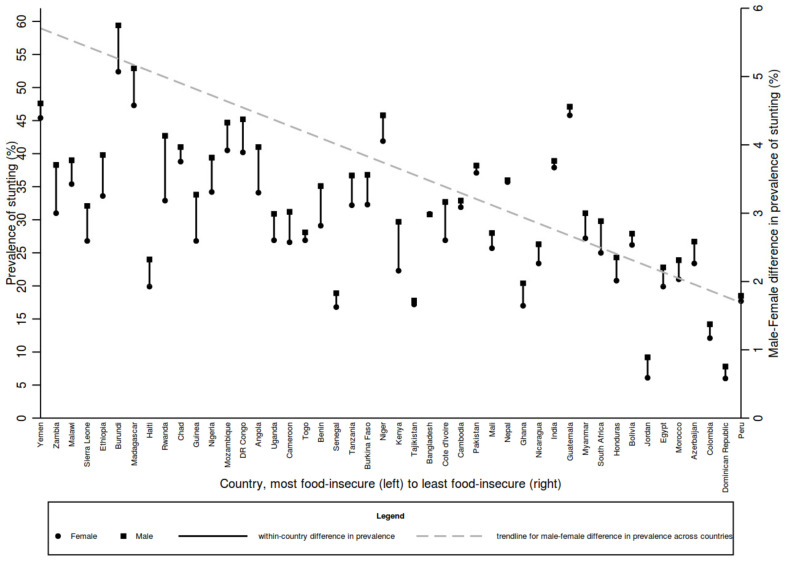
Prevalence of stunting in boys and girls by level of food insecurity. Data sources: Country food security scores from the global Food security index, found at https://foodsecurityindex.eiu.com/Index (accessed on 7 September 2021). Stunting prevalence data from the DHS StatCompiler https://www.statcompiler.com/en/ (accessed on 7 September 2021). The right *Y*-axis relates to the trendline for the male–female difference in prevalence across countries.

## Data Availability

Data used were from published studies. In addition to published studies, we sourced data from the global Food security index, found at https://foodsecurityindex.eiu.com/Index (accessed on 7 September 2021), and from the DHS StatCompiler https://www.statcompiler.com/en/ (accessed on 7 September 2021).

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
