# Peer review of "Understanding Sex Differences in Childhood Undernutrition: A Narrative Review"

_nutrients, 2022, doi:10.3390/nu14050948_

Round 1
Reviewer 1 Report
very nice paper
Author Response
Thank you so much for your positive feedback, we are very pleased you enjoyed the paper.
Reviewer 2 Report
This is an interesting manuscript that provides a discussion of the potential underlying factors explaining sex differences in undernutrition - including genetic factors, socio-economic status, infectious disease exposures, environmental exposures, and social preferences and practices. I appreciate how this manuscript builds upon a previous meta-analysis conducted by the authors and found their lines of inquiry to be solid. Specifically, I found the points that male fetuses have higher energy requirements/more susceptible to energy restrictions and that boys are introduced to complementary foods/higher vulnerability to infectious exposures - to be well argued. I have one major, and a few minor suggestions to improve the manuscript.
Major Suggestion
While I appreciated the main content of the manuscript (section 4.0) - I found the preceding section (section 3.0) to be too long and then not directly connect to section 4.0 (the explanations). It seemed that lines 92-96 (which explain the methodology for the earlier paper) and then section 3.0 (which present the results from the previous paper?) are too detailed for THIS paper. A quicker summary of the meta-analysis would allow the reader to jump into the main argument of this paper - the potential explanations of male vulnerability to undernutrition. While "major", I don't think this revision is difficult.
Minor Suggestions
4.4. Infant and young child feeding Infant feeding and care practices. ïƒ this heading is off
"The authors note however that the evidence for sex specificity in human milk composition is currently limited and conflicting but suggest that improved understanding of sex specific human milk composition may be essential in optimising nutritional care in early life and the male disadvantage." --- how so? Is the suggestion to supplement? What about the argument that supplementation creates exposures to pathogens?
The authors use the word "whilst"…. a lot.
Some of the spacing seems off.
Finally, I really appreciated the points about the limitations of current metrics. For example, the authors mentioned that LBW does not account for sex and issues with MUAC. It would be great to see this reiterated in the discussion.
Author Response
Thank you very much for your feedback, it is very much appreciated.
- We understand that section 3 was a little out of sync with the rest of the paper. The lines 92-96 do refer to methodology used in this review rather than the previous, but we agree with your overall point that the section could be edited to be less repetitive and fit better in the overall paper. We have integrated the section on prevalence into the introduction. The age and SES paragraphs have also been moved to within the main results section. We hope this helps to streamline the paper.
- We have amended the title in line 350
- We have removed the sentence on improved understanding of sex specificity in human milk
- We have removed some uses of the word “whilst”. Thank you for highlighting this.
- We have reviewed and edited the spacing
- Thank you for your comment on metrics. We have amended lines 534-536.

Reviewer 3 Report
This is an important and excellent paper, made stronger by the quantitative review published prior. I have only a few suggestions.
The abstract is a little frustrating to read...instead of presenting a selection or examples of some of the interesting findings from the narrative review, it states in sweeping generalizations that eidence on why sex differences occur, is limited but a number of factors contribute. especially general are the comments about infant and child feeding practices, environmental factors and the social and economic circumstances in which children are raised. This is a predictable and not very informative summary of some very interesting information imbedded in the paper. Is it possible to tantalize readers with some of these?
The one concern with the paper is that rarely are comments made about the methodological issues in measurement that likely contribute to conflicting results, or raise questions about the validity of results. This applies certainly to all the studies of infant and child feeding practices and attitudes, breast milk quantity and quality, and social norms re gender roles, etc. Preliminary comments about better methods for validity could be useful in the paper.
Spelling mistake..."separate" line 428.
Why mention sex differences in childhood obesity.,,line 549. A totally different issue.
Author Response
Thank you so much for your feedback.
- We agree with your comments about the abstract and have rewritten a little to reflect your recommendations.
- We have included a line in the discussion (lines 537-540) highlighting some of the challenges in sex not being the main focal point of many of the studies and how this limits overall what we can draw from this review
- We have corrected the spelling mistake you identified on line 428, thank you.
- We have removed the line on obesity

Round 2
Reviewer 2 Report
I appreciate the authors incorporating my recommendations into the text - looks good to go for me!